The effect of big data technologies usage on social competence

Elfeky Abdellah Ibrahim Mohammed abdalah.elfeqi@spe.kfs.edu.eg 1
Hassan Najmi Ali 2
Yasien Helmy Elbyaly Marwa 3
1 Department of Curriculum and Instruction, College of Education, Najran University , Najran , Saudi Arabia
2 Department of Educational Technology, King Abdulaziz University , Jeddah , Saudi Arabia
3 Centre for Sharia, Educational and Humanities Research, Najran University , Najran , Saudi Arabia
Alatas Bilal
Electronic publication date: 2023 Nov 17
Publication date: 2023
Volume: 9
Electronic Location ID: e1691
Received 2023 Jun 20; Accepted 2023 Oct 19
Copyright: ©2023 Elfeky et al.
Copyright year: 2023
Copyright holder: Elfeky et al.
License: This is an open access article distributed under the terms of the Creative Commons Attribution License, which permits unrestricted use, distribution, reproduction and adaptation in any medium and for any purpose provided that it is properly attributed. For attribution, the original author(s), title, publication source (PeerJ Computer Science) and either DOI or URL of the article must be cited.
License URL: https://creativecommons.org/licenses/by/4.0/

Keywords: Big data technology, Digital environments, Electronic social competence, Optimal investment, Big data

Funding: Deanship of Scientific Research at Najran University NU/DRP/SEHRC/12/13 The authors received funding from the Deanship of Scientific Research at Najran University for this work, under the General Research Funding program grant code NU/DRP/SEHRC/12/14. The funders had no role in study design, data collection and analysis, decision to publish, or preparation of the manuscript.

==============================
The learning management system is a digital environment that enables the tracking of learner activities, allowing special forms of data from the academic context to be explored and used to enhance the learning process. This study aims to identify the effect of using big data technology in digital environments on the development of electronic social competence among optimal investment diploma students. An experimental method was used to explore the effect of big data technologies usage on social competence. The sample for this study consisted of (120) students in the Department of Curriculum and Teaching Methods, divided into two equal groups through random selection. The first group studied the course through a digital environment with the use of big data technology, while the second group studied the course through the digital environment without using big data technology. The electronic social competence scale was further utilized as a tool to meet the study’s goal. The experimental findings showed that big data technologies in the used digital environment significantly improved the electronic social competence of Optimal Investment Diploma students (personal skills, self-management skills, and academic skills). The results provide significant proof of the advantages of big data technology in social competence studies and development.

Introduction

Social competence is a key quality for college students, as it increases their social cohesion and meets their needs as important members of the classroom, family, and community. Academic success and social competence are related, as improving social competence enhances academic success (Elbyaly & Elfeky, 2023; Tabassum, Akhter & Iqbal, 2020). Social competence is the ability to successfully achieve personal goals while using social behaviors that are appropriate for the social contexts in which they are used. This includes the ability to interact effectively with teachers and peers, navigate social situations, communicate effectively, manage emotions, adapt to different contexts, accept others, and self-evaluate one’s effectiveness in social interactions (Martinez et al., 2021; Saltz & Krasteva, 2022; Romera et al., 2017; Almalki & Elfeky, 2022). Social competence also helps students meet their long- and short-term developmental needs in social environments (Seitz, 2021). In the educational environment, social competence helps students to successfully access courses, meet associated personal, social, and emotional needs, and develop skills and attitudes that are applicable outside the educational institution (Elbyaly & Elfeky, 2023).

Recently, the term electronic social competence has surfaced with the shift towards e-learning in many higher education institutions using different educational technologies. With the development of e-learning platforms, electronic social competence in electronic environments has become increasingly important for social adaptation. Electronic social competence is defined as a balance of social skills that help the individual to communicate with others, achieve a kind of social acceptance, and the ability to perform behavior that achieves desirable results. This can be done by using the capabilities and tools of the participatory e-learning environment (Elfeky, 2017). Social competence is a broad and relevant field (Blanca, 2021), and it is related to the ability of learners to be able to effectively communicate and interact in the educational environment (Maharani & Usman, 2019). Learners with high electronic social competence are more integrated into the electronic environment, accepting and sharing knowledge, and have a desire and motivation for more sharing and learning. They are also more adaptive, open-minded, and integrated with digital learning elements compared to learners with low electronic social competence (Alharbi, Elfeky & Ahmed, 2022).

The acceleration of digital transformation and improvements in information and communication technologies (ICTs) mean that different sectors must respond more efficiently to new challenges (Abarca, Palos-Sanchez, & Rus-Arias, 2020; Mora-Cruz, Saura & Palos-Sanchez, 2022). As a result, there have been many changes in teaching approaches and instructional structures in educational institutions around the world. Analysis is a process through which data is transformed into knowledge that helps make informed decisions (Wang et al., 2018). Big data analysis technology is an essential tool that can be used to extract valuable conclusions from data (Elbyaly & Elfeky, 2023; Javed, Zeadally & Hamida, 2019; Kühn et al., 2018; Rahmani et al., 2021). Big data analysis technology is a tool that combines mathematics, statistics, computer science, and a specific applied field. Another developing area of research that can offer stakeholders, such as educators and students, some understanding of the learning process is that of big data technologies (Ahmed, Alharbi & Elfeky, 2022; Binsawad, Abbasi & Sohaib, 2022). This technology aims to accurately analyze data, enhance creative online learning, and draw conclusions in order to derive insightful knowledge (Elfeky & Elbyaly, 2019; Javed, Zeadally & Hamida, 2019).

The digital revolution in social sciences should be taken into consideration, as new methods of information exchange are emerging in digital ecosystems (Muniesa, Saura & Díaz-Garrido, 2021; Reyes-Menendez, Saura & Filipe, 2019). Big data usage in higher education also allows us to understand the academic challenges facing learners and defining strategies to address them (Kamupunga & Chunting, 2019; Zhou et al., 2023). It gives educational researchers the opportunity to use automated methods and technology to examine a complex educational phenomenon on a large scale (Daniel, 2019; Elbourhamy, Najmi & Elfeky, 2023). The objective of this research work is to contribute to a better understanding of the adoption of this technology and service (Palos-Sanchez, Reyes-Menendez & Saura, 2019). In addition, the use of an approach based on big data analytics to develop the online learning process improves the quality of learning by detecting the learner’s pattern and improving training courses (Elfeky & Elbyaly, 2021; Masada, 2017; Song et al., 2017). Interactions in learning environments can be effectively modeled and quantified by capturing the actions of students participating in online video lectures and then using machine learning algorithms to analyze the outcomes (Farhan et al., 2018; Saeed, Al Aghbari & Alsharidah, 2020). Student behaviors while engaging with video lectures are called video interaction events, which include slow watching, pausing, and back searching. Access to learners’ data has never before been so easy due to pervasive technology advancements in recent years (Blasco-Arcas et al., 2022).

Furthermore, detailed records of student behavior, performance, and other learning-related activities are collected through digital learning platforms (Aguilar, 2018; Saltz & Krasteva, 2022). Learning management systems are considered digital learning platforms that include web-based technologies and software applications used by students for accessing educational content (Ahmed, Alharbi & Elfeky, 2022; Masada, 2017). On this basis, the Blackboard learning management system has been adopted as a platform for managing educational resources and facilitating both synchronous and asynchronous interaction in all courses at Najran University (Elfeky & Elbyaly, 2021). Blackboard records learner interactions meticulously by documenting all learners’ activities (Granić & Marangunić, 2019; Song, Zhang & Duan, 2017). Lecturers can easily access detailed reports on learners’ activities for specific actions (Elfeky, Alharbi & Ahmed, 2022), such as video interactions and events (slow watching, pausing, and back searching). These reports include graphical representations that course instructors can review. The Blackboard system’s reports also display learners’ activity on a daily basis or for specific periods. Complex and unstructured learner interaction data can then be transformed into actionable information of value to higher education institutions (Cantabella, Martínez-España & Ayuso, 2019).

Big data analysis technology also has a set of properties that have been summarized in the code (7V), which include velocity, volume, veracity, variety, visualization, value, and variability (Ahad et al., 2021; Elfeky & Elbyaly, 2023). Additionally, the data provided by the Blackboard can encompass features of the “7Vs”, including velocity. Data on all learners are accumulated in real-time at a high velocity. The exponential growth of the data accumulation rate defines velocity, contributing to high processing costs and complexity (Duda, Kunanets & Matsiuk, 2018). Unlike traditional data warehouse analytics, which rely on periodic data loads and daily, weekly, or monthly updates, complex data volumes are analyzed and processed in real-time (Kumari, 2018). Volume refers to the continuous generation and compilation of records of all learners’ activities, resulting in a complex data volume. Blackboard data’s complex volume includes video interaction events (slow watching, pausing, and back searching). Veracity pertains to data consistency and accuracy (Duda, Kunanets & Matsiuk, 2018), which is relatively simple to achieve in the Blackboard platform since data originate from a single source, ensuring consistency and compliance. Variety characterizes the data types collected by the Blackboard platform, encompassing detailed statistics on interactive tools such as blogs, discussion forums, and video interaction events in unstructured or structured formats. Visualization, where the Blackboard platform is equipped with big data analytics techniques (as intelligent tools integrated into Blackboard), is utilized to generate reports for lecturers, who use them to make decisions on improving student learning. Such reports are based on algorithms and methods for real-time visual analytics (Kumari, 2018). Value implies the potential to generate economic value and new knowledge by leveraging the data. In other words, unstructured, complex data can be converted into actionable information and valuable insights using big data analytics techniques within the Blackboard platform, guiding decisions to enhance student learning. Variability refers to data whose meaning constantly changes (Choi, Ahn & Shin, 2019), such as video interaction data that can vary over time.

The rest of this article is organized as follows: a ‘Literature Review’ section, which provides a brief review of the relevant literature, is followed by the ‘Methodology’ section, presenting the approach adopted for the current study. The ‘Results’ section reports the findings, and finally, the article concludes with the ‘Discussion’, ‘Recommendations’, and ‘Conclusion’ sections.

Literature Review

Despite numerous research on using big data technology to improve teaching and learning (Elfeky & Elbyaly, 2017; Huda et al., 2018; Masadeh & Elfeky, 2016), there remains an insufficient focus on studying the impact of big data technology resulting from interaction in digital environments (Elfeky & Elbyaly, 2016; Shorfuzzaman et al., 2019; Wong, 2017). This gap in the literature may have implications for learning outcomes, particularly in terms of social competence. Consequently, our work aims to bridges this gap by examining the effect of using big data technologies in digital environments on the development of electronic social competence among optimal investment diploma students.

Najran University has adopted the Blackboard system as a digital environment for resource management and collaboration across all teaching methods and methods. This transformation has turned the educational community at the university into an electronic society, especially with the digital changes that accompanied COVID-19. In addition, online interaction and collaboration require improving students’ electronic social competence to enhance their ability to socialize in the new digital environment (Elbyaly, 2016; Reich, 2017). As we mentioned, improving social competence enhances academic success (Elbyaly & El-Fawakhry, 2016; Tabassum, Akhter & Iqbal, 2020). Moreover, upon reviewing the results of students’ tests in previous semesters, which indicated the low levels of achievement for a large percentage of students in the “Multimedia Programs” course. So, it becomes evident that there is an issue need to be addressed. Furthermore, none of the studies in the previous literature dealt with the study of variables related to this technology in digital environments to develop electronic social competence. This leads to the following research question:

RQ: What is the effect of using big data technology in digital environments on the development of electronic social competence among optimal investment diploma students?

Methodology

In this study, our goal was to determine the effect of using big data technology in digital environments on the development of electronic social competence among optimal investment diploma students. The research tool (Electronic Social Competence Scale) was introduced. Subsequently, steps have been taken to ensure the equalization of the two groups in terms of electronic social competence and the provision of experimental processing materials. Table 1 shows that the researchers employed the experimental method, utilizing a pre-post design with two equal experimental groups to explore the effect of using big data technology on electronic social competence.

Table 1 The experimental design.

	Pre-test	Treatment	Post-test	
The first group	Electronic Social Competence Scale	Big data technology in the digital environment	Electronic Social Competence Scale	
The second group	Digital environment		

Research tool (Electronic Social Competence Scale)

To evaluate the electronic social competence, the researchers develop a scale based on previous studies and educational literature that focused on measuring social competence, including works by Deptula et al. (2006), Gómez-Ortiz, Romera-Felix & Ortega-Ruiz (2017) and Zwaans et al. (2008). The electronic social competency scale consisted of three main dimensions: personal skills, self-management skills, and academic skills. These dimensions were also identified in previous studies and literature such as Alzahrani, Alshammary & Alhalafawy (2022) and Hocking et al. (2017).

To ensure the validity of the scale, it was also reviewed by a panel of experts and specialists. These experts were requested to assess the scale’s statements concerning their suitability for evaluating electronic social competence, the accuracy and clarity of language expressions, suggested additions or deletions, and any other observations or recommendations. Acceptance of a statement was determined if at least 80% of the experts and specialists agreed. The final version of the scale incorporated the observations provided by professionals and specialists, resulting in 26 statements. Each statement on the scale was rated on a five-point Likert scale, ranging from 1 (strongly disagree) to 5 (strongly agree). In addition, the scale was applied to a pilot sample of (23 students) not included in the main study. Moreover, Cronbach’s Alpha was used to assess the internal consistency of the paragraphs and the scale’s stability. The stability coefficient value (0.84) was determined for the scale as a whole. The average time taken by the first and last student to complete the scale was calculated, revealing that it took approximately (23 min) to administer. Thus, the scale was deemed suitable for assessing the electronic social competence of the study sample.

Ensuring equalization of the two groups in electronic social competence

The homogeneity of the two study groups was also confirmed by analyzing the extracted data through SEM structural equation modeling using multiple-group (CFA) dimensions of electronic social competence(personal skills, self-management skills, and academic skills).

Figure 1 shows that the electronic social competence in the first group was weakly affected by the personal skills dimension (p > .05, ß = .32), the self-management skills dimension (p > .05, ß = .24), and the academic skills dimension (p > .05, ß = .17). Likewise, electronic social competence in the second experimental group was weakly affected by each of the interpersonal skills dimension (p > .05, ß = .29), after self-management skills (p > .05, ß = .27), and after academic skills (p > .05, ß = .23). This indicates that the female students in both experimental groups were homogeneous and had equal abilities in electronic social competence before exposure to the experiment.

Figure 1 Multiple-group pre-CFA for the dimensions of electronic social competence for the two study groups.

The indicators of the suitability of the model are: CMIN = 194 (p = 0.382; CMIN (χ2)/df = .97; df = 198;), IFI = 93, PRATIO = .91, CFI.

Experimental processing material

Digital environments provide data about students’ interactions by keeping meticulous records of all of students activities (Cantabella et al., 2019). Administrators of digital environments have leveraged the advantage of big data analytics technology (as a smart tool) to generate reports that help lecturers in making decisions to improve students’ online learning. For instance, with activities like video interactions, such as slow watching, pausing, and back searching, lecturers can easily access comprehensive reports on specific students or the entire student body (Elbyaly & Elfeky, 2021).

The educational environment data was examined and saved in nine reports using this smart tool during the introduction of the “Multimedia Programs” course. These reports were carefully analyzed to determine their relevance to the study’s objectives. One of the selected reports was the “Student Overview of an Individual Course” displaying detailed statistics on video engagement events, including learners’ interactions with video lectures, arranged by date. The utilization of this report, in conjunction with the “tracking center” of the Blackboard system, facilitated the identification students who were at risk of failing in the course. In order to get the students’ attention and encourage them to actively participate in video lectures, the lecturer proactively engaged with the struggling students and provided immediate assistance using the tools available within the learning environment.

Ethical Statement

The Najran University Deanship of Scientific Research review board gave their approval (No.:444-45-22143-DS). The methods employed in this investigation adhere to the guidelines set forth in the Helsinki Declaration.

Consent form

We obtained informed written consent from all participants in our study.

Results

Post application scores for the electronic social competency scale dimensions (personal skills, self-management skills, and academic skills) were extracted for the two study groups. Then SEM using multiple-group CFA for statistical analysis and extraction of results.

Personal skills

The usage of big data technology in the digital environment had a significant and positive impact on the participants in the first experimental group’s dimension of personal skills, as demonstrated in Fig. 2 (p < .05, ß = .87). This dimension also had a significant positive effect on electronic social competence (p < .05, ß = .85). In contrast, the personal skills dimension of the learners in the second experimental group (p > .05, β = .42) had no significant effect on electronic social competence (p > .05, β = .33).

Figure 2 Multiple-group CFA for the dimensions of electronic social competence for the two study groups (Source: our own sources).

The indicators of the suitability of the model are: CMIN = 192 (p = 0.536; CMIN (χ2)/df = .98; df = 195;), IFI = 91, PRATIO = .90, CFI = 0.

Self-management skills

At the same time, the results showed that the dimension of self-management skills in the first experimental group was significantly affected via the use of big data technology in the digital environment (p < .05, ß = .88). This dimension also had a significant positive impact on the electronic social competence (p < .05, ß = .84). In contrast, the dimension of self-management skills in the second experimental group was weakly affected as a result of using the digital environment (p > .05, ß = .38). This dimension also had a weak effect on the electronic social competence (p > .05, ß = .37).

Academic skills

The usage of big data technologies had a significant impact on the academic skills dimension in the first experimental group (p < .05, ß = .91). This aspect also had a significant positive impact on electronic social competence (p < .05, ß = .88). In contrast, the use of the digital environment had little impact on the academic skills dimension in the second experimental group (p > 0.05, ß = 0.34). Additionally, electronic social competency was only marginally impacted by this aspect (p > .05, ß = .28).

Discussion

The study’s findings supported the idea that using big data technologies in a digital setting can help students strengthen their dimensions of electronic social competence (personal skills, self-management skills, and academic skills). These findings aligned with previous studies that have shown a positive relationship between big data technologies and different learning outcomes (Elfeky & Elbyaly, 2021; Qian et al., 2022; Sabaityte, Davidaviciene & Karpoviciute, 2020; Wei & Ren, 2022). They also agreed with other previous studies that examined the effect of using modern technologies and methods on social competence development. For example, Kim & Lee (2019), found that the amount of time children spend using smart media has a significant impact on their social skills, and Elfeky (2019) found that there is a primary effect of the difference in the level of cognitive control power on the development of electronic social competence in favor of students with high cognitive control power of the second order (who studied with the electronic discussion strategy based on the Blackboard system). However, the results of this study do not agree with the results of the study by Savelchuk et al. (2021), which found that social competence decreased during the COVID-19 pandemic. This suggests that the use of big data technologies may be more effective in promoting social competence in a face-to-face learning environment than in a distance learning environment. In summary, these results suggest that big data technologies can enhance the use of faculty members for the advantages of additional digital environments.

Conclusion and Recommendations

In this study, the Blackboard platform data was examined and saved in nine reports using big data analytics technology during the introduction of the “Multimedia Programs” course. These reports were carefully analyzed to determine their relevance to the study’s objectives. One of the selected reports was the “Student Overview of an Individual Course” displaying detailed statistics on video engagement events, including learners’ interactions with video lectures, arranged by date.

The utilization of this report, in conjunction with the “tracking center” of the Blackboard platform, facilitated the identification students who were at risk of failing in the course. In order to get the students’ attention and encourage them to participate actively in video lectures, the lecturer proactively engaged with the struggling students and provided immediate assistance using the tools available within the learning environment.

The electronic social competency scale was further utilized as a tool to achieve the study’s goal. The experimental results showed that the development of electronic social competence, encompassing personal skills, self-management skills, and academic skills, was significantly improved by big data technology in the digital environment.

Our results provide meaningful practical implications for faculty members looking for advantages of additional digital environments. In additions, the study importance is directing educators’ attention to the importance of overcoming educational difficulties facing beneficiaries in digital environments using big data analytics technology. Based on the results of the current study, the researchers recommended that educators should be interested in developing the abilities of learners with advanced technologies. In addition, they requested that those in charge of the educational process work to boost students’ abilities to live in harmony with one another in the new electronic society, and that further studies be conducted to explore the possibility of developing electronic social competence through augmented reality.

Theoretical Contributions

The current research contributes to the existing literature of big data technology in digital environments and social competence, as the development of electronic social competence was significantly improved by big data technology in the used digital environment. These results align with numerous earlier studies such as Elfeky & Elbyaly (2021), Qian et al. (2022), Sabaityte, Davidaviciene & Karpoviciute (2020) and Wei & Ren (2022).

Practical Implications

The current research also contributes to the existing literature on personal, self-management, and academic skills. Specifically, the use of big data technology in digital environments had a positive and significant influence on the development of these skills. The findings provide important evidence of the benefits of big data technology in skills development. These results also align with numerous earlier studies, including those conducted by Elfeky (2019) and Kim & Lee (2019).

Limitations and Future Directions

Even though the research was rigorous and thorough, there were certain limitations. Firstly, this research project focused on using big data technology in digital environments inside Saudi Arabia. Additional research with this technology is required in other countries or regions to assess its applicability and effectiveness in diverse educational settings. Secondly, while this research discovered positive results regarding the role of big data technology in digital environments, more research is needed to investigate the possibility of enhancing electronic social competence through other emerging technologies exploring the potential impact of technologies like artificial intelligence, virtual reality, or augmented reality on social competence development could provide valuable insights for educators and researchers. Finally, it is important to acknowledge that the sample size used in this study is limited. Conducting more extensive research with a larger and more diverse participant pool, encompassing various fields of study and institutions, could strengthen the generalizability and validity of the findings.

Supplemental Information

Data S1 Raw Data

Click here for additional data file.

Supplemental Information 2 Electronic Social Competence Scale

Click here for additional data file.

Additional Information and Declarations

Competing Interests

Author Contributions

Ethics

Data Availability

The authors declare there are no competing interests.

Abdellah Ibrahim Mohammed Elfeky conceived and designed the experiments, performed the experiments, prepared figures and/or tables, and approved the final draft.

Ali Hassan Najmi conceived and designed the experiments, analyzed the data, performed the computation work, authored or reviewed drafts of the article, and approved the final draft.

Marwa Yasien Helmy Elbyaly performed the experiments, prepared figures and/or tables, authored or reviewed drafts of the article, and approved the final draft.

The following information was supplied relating to ethical approvals (i.e., approving body and any reference numbers):

The Najran University Deanship of Scientific Research review board

The following information was supplied regarding data availability:

Raw data are provided as a Supplemental File.

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
