# Peer review of "The effect of big data technologies usage on social competence"

_PeerJ Computer Science, doi:10.7717/peerj-cs.1691_

## Round 0.1 · original submission · Major Revisions

Dear authors, after carefully reading the reviewer comments i have made the decision to give this paper a chance. However there are major improvements that need to be done in order to justify publication. If you consider you can implement them, accept to continue in the process and address each of the reviewer concerns.

**Language Note:** PeerJ staff have identified that the English language needs to be improved. When you prepare your next revision, please either (i) have a colleague who is proficient in English and familiar with the subject matter review your manuscript, or (ii) contact a professional editing service to review your manuscript. PeerJ can provide language editing services - you can contact us at copyediting@peerj.com for pricing (be sure to provide your manuscript number and title). – PeerJ Staff

·

Basic reporting

First of all, I would like to thank the journal and the authors of the research article for giving me the opportunity to have the privilege of conducting the following review.
The objective of the scientific study to be reviewed is to identify the use of technology to improve the learning process, in a digital environment.
I would like to start the review with the first suggestion of change regarding the TITLE of the article: Currently the title is "The effect of the use of big data technology in digital environments on the development of electronic social competence" a title that is too long without be too concise. I suggest the following change: "The effect of the use of big data technologies on social competence" to be understood more clearly by the rest of the researchers.
I consider the KEYWORDS chosen for the development of the research to be correct, since it is advisable to have a minimum of three keywords and a maximum of five to be optimal.
Regarding the ABSTRACT, its dimension is correct, but it is recommended that apart from the objective and the results, the methodology used be named and not only its development, as indicated at the beginning of the sentence "The sample for this study was divided into two groups by random selection of students...”
THE INTRODUCTION is one of the sections of the paper where more changes are made for its improvement: First, it is suggested to change the starting order of this section, that is, the third paragraph where the explanation of social competence is indicated should be at the beginning of in such a way that it put other researchers in context at the beginning of the reading and then the explanation of what Big Data is. The explanation of Big Data, it is recommended, that it be supported by definitions and theories of other authors and other research, through citations according to the APA model, for example, I suggest a series of interesting investigations related to the subject of the study:
- - -
In addition, authors are requested to include in the last paragraph of this section the structure that the article will follow. Regarding the structure of the work, the authors are requested to reorganize the information into the large blocks of 1) Introduction, 1) Theoretical framework or Literature review, 3) Materials and methods or Methodology, 4) Data analysis or Results, 5) Discussion, 6) Conclusions. Regarding the hypothesis questions, it would be convenient to ask two research questions for this study, that is, what questions does the study answer?
The methodology section develops the method used well, but it is suggested that the explanation and information of the same be expanded at the beginning of the section since it is not well understood what it refers to.
Reyes-Menendez, A., Saura, J. R., & Filipe, F. (2019). The importance of behavioral data to identify online fake reviews for tourism businesses: A systematic review. PeerJ
Computer Science, 5, e219.
Palos-Sanchez, P., Reyes-Menendez, A., & Saura, J. R. (2019). Modelos de Adopción de Tecnologías de la Información y Cloud Computing en las Organizaciones. Información
tecnológica, 30(3), 3-12.
Blasco-Arcas, L., Lee, H. H. M., Kastanakis, M. N., Alcañiz, M., & Reyes-Menendez, A. (2022). The role of consumer data in marketing: A research agenda. Journal of business
research, 146, 436-452.

Regarding the discussion, it must be supported by studies by other authors (with APA citation) that respond to the same theory exposed in the article, if not, justify it.
The conclusions and recommendations are suggested to the authors to join both sections, leaving a clear explanation of the theoretical and practical implications of the research method, as well as the results obtained.
The tables and graphs of the paper must always be accompanied by a title and source, in the case of having been prepared by the authors, indicate that they are their own sources.
The research topic presented by the authors is interesting, but it requires the suggested improvements and the updating of the data presented.

Experimental design

Introduction
Discussion
Conclusion

Validity of the findings

No comment

Additional comments

No comment

Cite this review as

·

Basic reporting

Thank you very much for the opportunity to read your manuscript entitled: “The effect of using big data technology in digital environments on the development of electronic social competence” to PeerJ Computer Science Journal.
The purpose of this study is to identify the effect of using big data technology in digital environments on the development of electronic social competence among optimal investment diploma students.
The following are my comments, all intended to improve the final result of this research work.
The focus of the paper
Overall, this is an interesting research topic, closely related to the electronic social competences.

Abstract
The abstract presents the main research objectives and methodology; however, the methodology is not shown completely (lack of type, name…, survey); the research findings and implications are not clear.

Introduction
- The paper's structure is not shown in the introduction.
Theoretical part

- The authors should add findings in terms of topics, methods,... related the electronic social competences that the authors must summarized from the systematic literature reviews located.
-Add recent references such as:
Mora-Cruz, A., Saura, J. R., & Palos-Sanchez, P. R. (2022). Social media and user-generated content as a teaching innovation tool in universities. In Teaching innovation in university education: Case studies and main practices (pp. 52-67). IGI Global. DOI: 10.4018/978-1-6684-4441-2.ch004
Palos-Sanchez, P., Saura, J. R. & Velicia-Martin, F. (2022). A case study on a hedonic-motivation system adoption model in a game-based student response system. International Journal of Human-Computer Interaction. DOI: https://doi.org/10.1080/10447318.2022.2121801
Abarca, V. M. G., Palos-Sanchez, P. R., & Rus-Arias, E. (2020). Working in virtual teams: A systematic literature review and a bibliometric analysis. IEEE Access, 8, 168923-168940. doi: 10.1109/ACCESS.2020.3023546

Experimental design

The aim, Methodology, and Data
- Research aims and research questions were not presented in the Introduction section and specific statements on the research novelties are not well-explained.
- Description of the main research method (study area, method…? Survey?) is not clear and appropriate.
- The author should also mention which method is used to answer which specific research question (among 4 questions).
Results
- Explain with more details figures and tables.

Validity of the findings

Discussion
There is no link between the result part (mainly based on the data survey ) with the discussion part (mainly based on the literature review). Therefore, it seems that the results obtained from the survey study do not provide interesting discussions or implications.
Conclusion
- The conclusion is too general, failing to summarize the highlights of the results and discussion, especially the survey results.
- The authors should add the limitation of the research/results and directions for future research
References
Add DOI for all references.
Formal requirements
This paper is in need of proofreading.

Cite this review as

---

## Round 0.2 · Minor Revisions

Thank you for the revision of your work. The reviewers consider that you have conducted a successful improvement of your research with their comments
However, I still have some concerns as to how appropriate the research is for this journal PeerJ Computer Science

I will therefore ask for a minor revision in order to reinforce the suitability of the topic for this journal

Also, take the chance to improve the conclusions making them innovative and not repeating the previous implications and results

·

Basic reporting

ok

Experimental design

ok

Validity of the findings

ok

Cite this review as

---

## Round 0.3 · accepted · Accept

Dear authors,

The original editor was unavailable so I took over the editorship of your paper.

Thank you for the revision. It appears that all of the reviewers' comments have been clearly addressed. Your article is accepted for publication after the last revision.

Best wishes,